# The Curse of Conservation: Empirical Evidence Demonstrating That Changes in Land-Use Legislation Drove Catastrophic Bushfires in Southeast Australia

Alice Laming [1], Michael-Shawn Fletcher [1,2,3,*], Anthony Romano [1], Russell Mullett [4] on behalf of Gunaikurnai Land and Waters Corporation, Simon Connor [3,5], Michela Mariani [3,6], S. Yoshi Maezumi [7] and Patricia S. Gadd [8]

1  School of Geography, Earth and Atmospheric Sciences, The University of Melbourne, Carlton, VIC 3053, Australia
2  Indigenous Knowledge Institute, The University of Melbourne, Parkville, VIC 3010, Australia
3  Australian Research Council Centre of Excellence for Australian Biodiversity and Heritage, University of Wollongong, Northfields Ave, Wollongong, NSW 2522, Australia
4  Gunaikurnai Land and Waters Aboriginal Corporation (GLAWAC), Kalimna West, VIC 3909, Australia
5  School of Culture, History and Language, The Australian National University, Canberra, ACT 0200, Australia
6  School of Geography, University of Nottingham, Nottingham NG7 2QL, UK
7  Department of Archaeology, Max Planck Institute for the Science of Human History, 80539 Munich, Germany
8  Australia's Nuclear Science and Technology Organisation, Lucas Heights, NSW 2234, Australia
*  Correspondence: michael.fletcher@unimelb.edu.au

**Abstract:** Protecting "wilderness" and removing human involvement in "nature" was a core pillar of the modern conservation movement through the 20th century. Conservation approaches and legislation informed by this narrative fail to recognise that Aboriginal people have long valued, used, and shaped most landscapes on Earth. Aboriginal people curated open and fire-safe Country for millennia with fire in what are now forested and fire-prone regions. Settler land holders recognised the importance of this and mimicked these practices. The Land Conservation Act of 1970 in Victoria, Australia, prohibited burning by settler land holders in an effort to protect natural landscapes. We present a 120-year record of vegetation and fire regime change from Gunaikurnai Country, southeast Australia. Our data demonstrate that catastrophic bushfires first impacted the local area immediately following the prohibition of settler burning in 1970, which allowed a rapid increase in flammable eucalypts that resulted in the onset of catastrophic bushfires. Our data corroborate local narratives on the root causes of the current bushfire crisis. Perpetuation of the wilderness myth in conservation may worsen this crisis, and it is time to listen to and learn from Indigenous and local people, and to empower these communities to drive research and management agendas.

**Keywords:** south-east Australia; fire; indigenous land management; conservation; wilderness; fuel; cultural burning; British invasion; Anthropocene

## 1. Introduction

Society is placing ever-increasing pressures on the Earth system. This has led to the recognition of a new geological epoch, the Anthropocene, to draw attention to the widespread impacts associated with post-industrial human activity [1–5]. The conservation movement of the 1950–1970's, is often heralded as an antidote to the detrimental environmental impacts of modernity [6–9]. This movement saw the establishment of the wilderness act in the USA in 1964, the first of a series of enacted pieces of legislation through the subsequent decades that sought to offer protection for the remaining wilderness areas across many parts of the Earth [10–12]. Wilderness-inspired conservation ideology is based on the notion that much of the Earth was free from human influence prior to the Industrial Revolution and the over-exploitation of our environment that was required to fuel it [13–16]. Despite a long history of critique-including empirical data demonstrating that little (<20%) of the Earth has avoided human influence for more than 12,000 years [17], the

wilderness ethos still underpins many approaches to environmental management across the globe [16,18].

In Australia, a land that has been inhabited and managed by people for over 65,000 years [19], it has been argued that more than 40% of the landscape is comprised of wilderness [20]. Indeed, Australia hosts the most national parks of any country on Earth and the second largest area of national park reserves (behind Canada) [21]. The enormous National Park estate, and other government-managed reserves in Australia, are largely removed of human influence and are regarded as important sanctuaries for biodiversity on a continent that is experiencing the second-fastest rate of biodiversity loss on Earth [22–24]. The increased frequency and intensity of bushfire, invasive species, habitat loss/fragmentation and climate change are seen as the key drivers of the alarming biodiversity loss in Australia ("bushfire" is a commonly used term in Australian fire discourse and is equivalent to the term "wildfire" used elsewhere) [25–27]. This narrative underpins a strong push from the conservation sector to annex and protect more areas of what are considered the last remaining wildernesses [28,29]. In contrast, little attention is paid to the potentially harmful impacts of imposing this style of wilderness-inspired conservation on landscapes long managed by people [16,30–32].

Land management is performed by all people to promote a safe, predictable and resource rich environment. Aboriginal approaches to land management in Australia are local-scale, complex and varied, and are based on reciprocal relationships with Country and are more aptly described as care, rather than management [33–37]:

> *"Country is multi-dimensional—it consists of people, animals, plants, Dreamings; underground, earth, soils, minerals and waters, air . . . People talk about country in the same way that they would talk about a person: they speak to country, sing to country, visit country, worry about country, feel sorry for country, and long for country."* [38].

The systematic and sophisticated application of fire to Country—based on local knowledge, kinship and cultural protocols that developed over thousands of generations of knowledge accumulation and sharing—is known as "cultural burning" [37,39,40]. Cultural burning, while spatially and temporally heterogeneous and specific to the type of Country and local cultural protocols, has the net effect of creating and maintaining productive and diverse landscapes, with an overall lower woody biomass and higher grass and herb cover [37,41–44]. Viewed from a bushfire management perspective, cultural burning acts to suppress wood and shrub fuel loads overall, reducing problematic ladder fuels (which connect ground fires to the canopy/crown layer) and lowers the risk of catastrophic crown fires [43,45,46]. Early colonists and settlers in southeast Australia recognised the profound benefits of cultural burning on Country, both in terms of risk management and pasture management, and many replicated the approach on the land stolen from the traditional Aboriginal owners (we hereon refer to the practice of settlers emulating cultural burning as "settler mimicry") (n.b. our definition of "settler mimicry" does not include hazard reduction burning, which is a form of fire suppression focused solely on fuel reduction and explicitly targeted around high priority assets, such as property and human settlements [47–49].

Here, we set out to investigate how changing management approaches since the early 1900s in response to conservation/wilderness-inspired legislation influenced fire regimes in part of southeast Australia prone to catastrophic bushfires. We present data on changes in vegetation, fire and erosion spanning the past 120 years from a place now known as Buchan on Gunaikurnai Country (Gippsland) in southeast Australia. Gunaikurnai Country includes coastal and inland areas, alpine vegetation, forests, and remnant grassy plains. Much of Gunaikurnai forest estate is regarded as wilderness by government agencies and conservation groups, and large areas are contained within the National Parks system [50,51]. This region has experienced several catastrophic bushfires over recent decades, including the 2019–2020 "Black Summer" bushfires—the largest bushfires on record in Australia (>24 million hectares) [52,53]. The Buchan region is comprised of large areas of dense Eucalypt-forest in a temperate climate that receives 800–1200 mm of rainfall annually.

These dense and highly flammable forests carried some of the most intense fires during the 2019–2020 Black Summer event [54–57]. We analysed sedimentary pollen (an indicator of vegetation change), charcoal (an indicator of changes in fire) and magnetic susceptibility (an indicator of soil erosion) from a small billabong (ox-bow lake) measuring 1800 m$^2$—Tooculerdoyung Lagoon (37°31′00″ S, 148°15′43″ E)—on the banks of the Snowy River, near the confluence of the Snowy and Buchan Rivers (Figure 1c,d)—Tooculerdoyung is a Gunaikurnai word that translates as—"a point of river". The site is currently surrounded by dense temperate eucalypt-forest and lies in the center of a well-known lithograph drawn by renowned realist painter Eugene von Guérard (painted in 1867) in which the landscape surrounding the site is depicted as an open forest (Figure 1f).

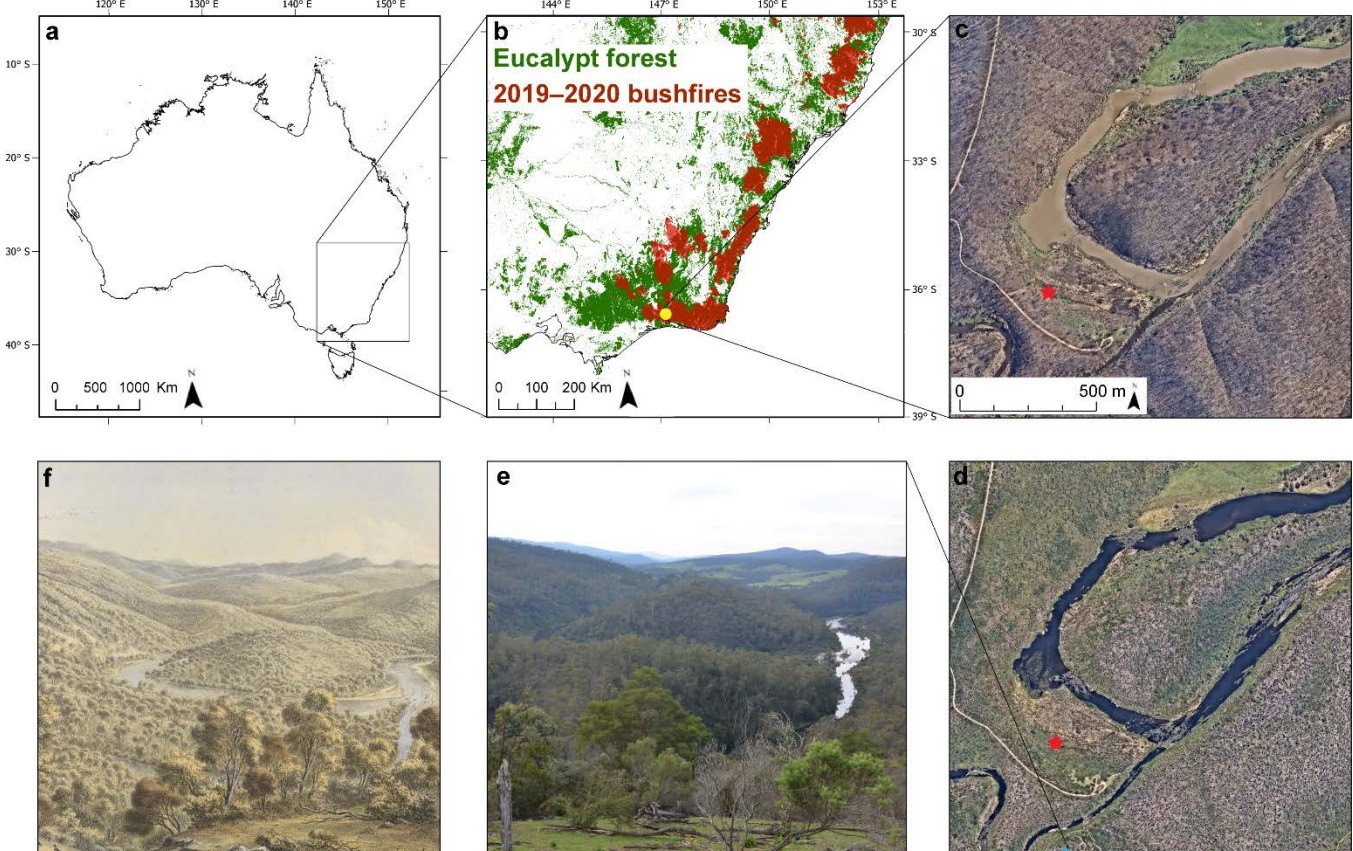

**Figure 1.** (**a**) Map of Australia, black square indicates study region; (**b**) Southeast Australia study region with Eucalypt forests (green) [58], the 2019–2020 Black Summer bushfires burn area (red) [59] and the location of the site (yellow dot); (**c**) Satellite image of the confluence of the Snowy and Buchan Rivers and the study site (red star) after the Black Summer bushfires (taken 27th February 2020; [60]); (**d**) Satellite image of the confluence of the Snowy and Buchan Rivers and the study site (red star) with forest recovering (taken 14th December 2020; [60]); (**e**) Photograph of Junction of the Buchan and Snowy Rivers taken by Professor Bruno David (taken 2019, shortly before the 2019–2020 bushfires); (**f**) Junction of the Buchan and Snowy Rivers, Gippsland (painted in 1867) colour lithograph, Eugène von Guérard [61]. For interpretation of the references to colour in this figure, the reader is referred to the online version of this article.

We aim to address the following research questions: (1) is there empirical evidence to support the landscape openness depicted in the ethnopictoral record? (2) Have changing management approaches impacted the vegetation and fire regime of the area? (3) What was the impact of imposing conservation/wilderness protection ideals on this landscape in the mid-1900s?

## 1.1. Study Region

Southeast Australia is host to high biomass temperate broadleaf forests and shrublands dominated by species of the eucalypt genus, as well as grassland. Eucalypts are among the most flammable trees on Earth with dense populations contributing to high fuel loads and crown fires [44,62–65]. Fire activity at the landscape scale is determined by three key factors: 1 ignition (humans and lightning); 2 fuel (biomass); and 3 climate (via its influence over fuel dryness) [66]. Analysis of fire activity over the recent past reveals climate (El Niño–Southern Oscillation [ENSO]) as the principal control over inter-annual fire activity across southeast Australia [67]). Prior to the British Invasion, forested landscapes had lower woody/shrubby fuel-loads and fewer bushfires than during the subsequent post-Invasion period [41]. The recent increase in landscape fuel and shift in fire dynamics occurred in response to the removal and suppression of Aboriginal cultural burning imposed by the British following their Invasion of Aboriginal lands [34,44]. Early descriptions of southeast Australia by British settlers and landscape artists depict the region as largely open and grassy (e.g., in Figure 1f). The open and grassy nature of the southeast Australian landscape at the time of the British Invasion and the promise of the "richest pastures in the world" was an inspiration that spurred the British to colonise the continent [68]. Today, much of the region outside of agricultural and urban zones is covered by thick and shrubby eucalypt forest (Figure 1e)

## 1.2. A confluence of Factors

The modern global conservation movement that began in the early 1900s and revolved around three central tenets: (1) human activity was harmful to the environment; (2) it is our duty to protect the environment from human-caused harm for future generations; and (3) that science and empirical data are needed to understand the problem and execute this duty [7,8,69–71]. This separation of humans from the "natural" world and the prioritisation of science over other ways of knowing paved the way for the denial of human agency in the Australian landscape by the British (i.e., the myth of *terra nullius* that legalised the theft of Aboriginal land by the British). This ultimately led to the outright neglect of parts of Country by the new British occupants who deemed such areas as unsuited for agrarian or other purposes [16,71–74]. Indeed, the use of the term "wilderness" increased across the English-speaking world following the "discovery" of what is now known as Australia by the British in the late 1700s (Figure 2). Despite the efforts of the early modern conservation movement, environmental destruction resulting from the continued European colonisation of Aboriginal lands and European extractive approaches to the natural world continued unabated.

The increased demands of industry and consumption through the 20th century, coupled with ever-increasing impacts of disasters such as oil spills, pesticides use, and industrial dumping in areas considered pristine "wilderness", saw a resurgence in the use of this term (Figure 2) [7,8]. This also drove a wave of conservation legislation that sought to protect areas considered free of human impact: i.e., wilderness areas [7,8]. The USA Wilderness Act of 1964 (Figure 2), for example, encompassed 800 designated wilderness areas to be managed by the National Park Service and other government bodies [7,8,13]. In Australia, this wave of wilderness-inspired conservation legislation culminated in acts such as the Victorian Land Conservation Act in 1970 and the federal National Parks Act in 1975 (Figure 2), advocating for the protection of Crown land for enjoyment, recreation and education of the public [75,76].

These land protection acts sought to protect areas that were perceived as free from destruction at the hands of people. In Victoria, where the present study is based, the Land Conservation Act of 1970 was preceded by the Forest Act (1958) (Figure 2), which legislated a fire-suppression approach to fire management in Victorian forests [77] (S. 61A onwards). Together, these Acts aimed at preserving people-free nature and advocated for complete removal of human agency and the active suppression of fire in landscapes that had been previously managed/cared for by Aboriginal people with fire (among other approaches) for

millennia. There has been a shift toward the application of fuel reduction within forests systems in Victoria over more recent years, such as hazard reduction burning [57,78]. Despite this shift, under-resourcing of forest management agencies and the degree of post-British Invasion forest expansion and thickening have rendered these methods largely ineffective at mitigating catastrophic bushfires under extreme fire weather conditions [41,79–81].

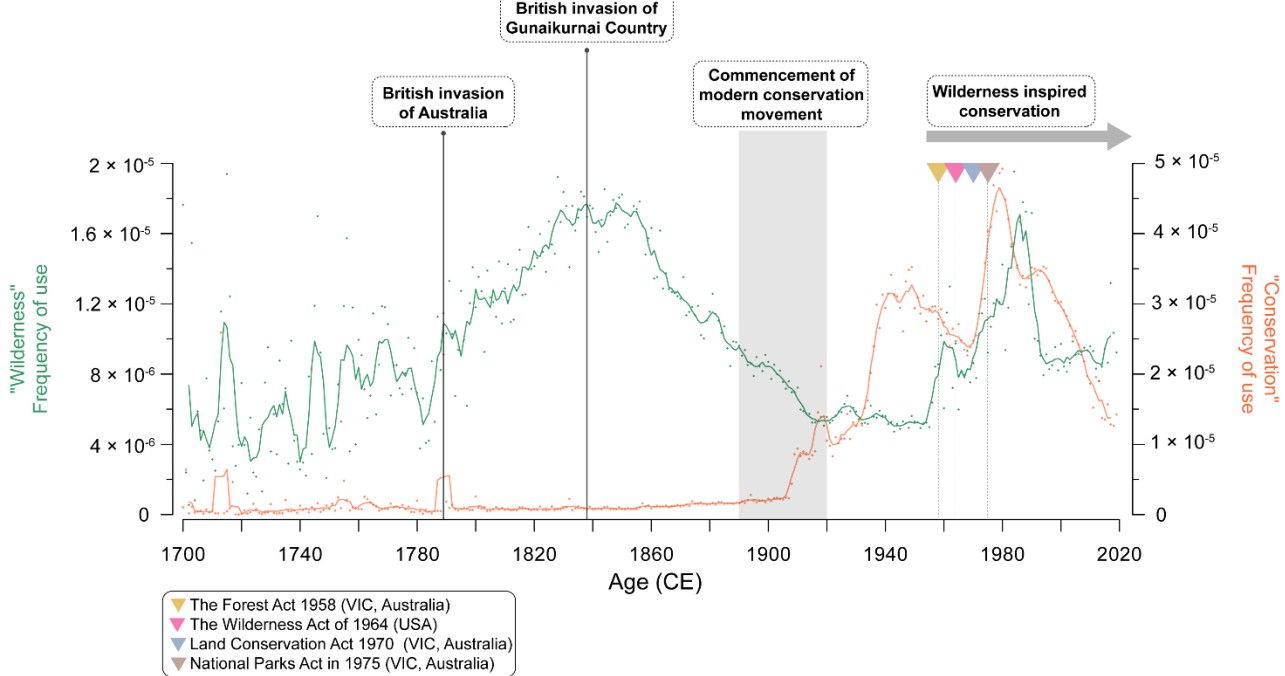

**Figure 2.** The frequency of the use of the terms 'wilderness' (green) fit with a weighted average (window width = 5) (solid green line) and 'conservation' (orange) fit with a weighted average (window width = 5) (solid orange line) from 1700–2019. Data derived from Google Books Ngram Viewer using ngramr in R [82]. This study searched the terms (ngrams) 'wilderness' and 'conservation' (not case sensitive) in books published between 1700 and 2019 within the digitised 'English 2019' (eng_2019) corpus (all books in the English language published in any country available on Google). The 2019 corpus is the most up to date with more books, improved tokenisation, optical character recognition (OCR) and library/publisher metadata. Ngrams were searched when they also crossed page boundaries but not across sentence boundaries. For interpretation of the references to colour in this figure legend, the reader is referred to the web version of this article.

On Gunaikurnai Country, the British Invasion and subsequent colonisation and settlement began by 1839, at a time when thousands of Gunaikurnai people were living on their tribal lands. Within 20 years of Invasion, and following the brutal massacres by Angus McMillan and others, only 100 people were thought to be surviving [83–85]. Upon arrival on Gunaikurnai land, settlers made note of "very open forests and visibility [was] possible for more than a mile in many places" [86]. Efforts to keep the area open were employed through ringbarking (the process of killing a tree by cutting a ring around it deep enough to sever the movement of water and nutrients through the tree) and settler mimicry burning [47]. Despite the settler mimicry employed in parts of the landscape deemed economically viable, much of the landscape was neglected and a rapid regrowth of woody plants followed, including in some areas that the British abandoned. This abandonment enabled forest to replace more open vegetation [86]. A consequence was a series of catastrophic bushfires across highly flammable and unmanaged Country [47]. The outcomes of a review in to the causes of the repeated catastrophic bushfires on Gunaikurnai Country between 1939–2007 identified that areas managed with low-intensity frequent fires by settlers experienced lower fire severity during the 1939 bushfire (one of the most severe fires

recorded in southeast Australia during the historical period [47,87]). Economic pressures through the mid- to late-1900s saw a shift from numerous small land-holder farmers to fewer big holdings who managed the land at much larger scales than Gunaikurnai or the settler farmers who displaced them [86,88]. This shift in the scale of management occurred at the same time as the establishment of the Land Conservation Council (under the Lands Conservation Act) which, among many sweeping changes, sought to minimise human impacts on landscapes and actively prohibited settler mimicry burning [76,89].

The impact of this constellation of factors—including the British Invasion, wilderness-inspired conservation legislation, bushfire suppression, shifting scale of landscape management—had profound impacts on the landscape configuration of Gunaikurnai Country. In the aforementioned 2007 Inquiry into the impact of public land management practices on bushfires in Victoria, local knowledge holder Gilbert Rothe stated that once appreciated national parks are now " ... being eroded by an overgrown, inaccessible floor cover of debris" (i.e., drastically increasing fuel loads since the enactment of such legislation) [47,87]. Rothe explicitly implicates the prohibition of settler mimicry burning by the Land Conservation Council from 1970 in allowing forests to become overloaded with fuels, rendering them a catastrophic fire risk:

> *"In our area it was 35 years ago when the use of the burnt areas for cattle grazing was stopped ... That happened in about 1970, and in fact it was when the Land Conservation Council first started that it stopped cattle grazing and took the runs off us in our area. That is when the demise of all this started happening. The older ones who are still around will still tell you that one day they will burn us out, because there is no management in the bush anymore as far as fire suppression goes, and, really, it is coming true."* [87]

Here, we set out to investigate this notion of the impacts of changing management approaches since the early 1900s and the impact of conservation/wilderness-inspired legislation on Aboriginal landscapes that are vulnerable to large-scale catastrophic fires when neglected

## 2. Materials and Methods

### 2.1. Core Collection & Chronology

An 86-cm sediment core (VIC2104A) was extracted from the deepest and center-most point in Tooculerdoyung Lagoon (Figure 1c,d) using a Universal Corer [90]) in January 2021. Tooculerdoyung Lagoon is located on Gunaikurnai Country. Permission to collect and analyse the sediment was provided by local Gunaikurnai Elders and the Gunaikurnai Land and Waters Corporation (GLAWAC). Samples for [210]Pb dating were chemically processed and analysed by alpha spectrometry [91] at the University of Ottawa (sixteen [210]Pb samples) (Table 1). All [210]Pb samples were bulk sediment dried in a convention oven prior to analysis. The Constant Rate of Supply (CRS) model [92]) was used for calibrating [210]Pb dates. An age-depth model was developed in R v. 4.2.0 [93] using the Clam package v. 2.4.0 [94]

**Table 1.** Lead-210 results and Constant Rate of Supply (CRS) model ages.

| Depth (cm) | Cumulative Dry Mass (g/cm$^2$) | Total 210Pb (Bq/kg) | Unsupported 210Pb Decay (Bq/kg) | Calculated CRS Age (as Calendar Years CE) | CRS Error (Years) |
|---|---|---|---|---|---|
| 0 | 0 | | | 2021.09 | 0 |
| 0.25 | 0.04 | 157.86 | 4.56 | 2020.24 | 0.07 |
| 0.75 | 0.14 | 138.62 | 3.80 | 2018.24 | 0.17 |
| 2.25 | 0.25 | 126.20 | 4.62 | 2016.16 | 0.24 |
| 2.75 | 0.37 | 108.04 | 3.61 | 2014.01 | 0.31 |
| 3.25 | 0.51 | 100.99 | 4.82 | 2011.65 | 0.39 |

| Depth (cm) | Cumulative Dry Mass (g/cm$^2$) | Total 210Pb (Bq/kg) | Unsupported 210Pb Decay (Bq/kg) | Calculated CRS Age (as Calendar Years CE) | CRS Error (Years) |
|:---:|:---:|:---:|:---:|:---:|:---:|
| 5.75 | 0.66 | 81.58 | 3.89 | 2009.31 | 0.46 |
| 7.25 | 0.84 | 85.51 | 9.94 | 2006.44 | 0.58 |
| 10.25 | 1.07 | 98.07 | 5.21 | 2001.74 | 0.74 |
| 15.25 | 1.27 | 110.20 | 5.53 | 1996.58 | 0.90 |
| 20.25 | 1.44 | 107.80 | 8.72 | 1991.10 | 1.07 |
| 30.75 | 1.65 | 67.99 | 5.78 | 1984.68 | 1.31 |
| 40.75 | 1.93 | 45.18 | 3.92 | 1977.73 | 1.61 |
| 55.75 | 2.28 | 59.00 | 6.00 | 1967.10 | 1.88 |
| 70.75 | 2.65 | 43.28 | 4.41 | 1950.75 | 2.56 |
| 81.75 | 3.01 | 55.52 | 4.48 | 1918.21 | 3.39 |
| 85.75 | 3.41 | 30.44 | 1.90 | | |

### 2.2. Pollen

Sub-samples of 0.5 cm$^3$ were taken every 3 cm between 0–86 cm and every 2 cm between 12–24 cm (32 pollen samples) for pollen analysis. Pollen was prepared according to standard methods adapted from procedures in Faegri and Iversen [95]. A minimum of 300 terrestrial pollen grains were counted for each sample. Where *Eucalyptus* spp. counts were >100, counting continued until 200 non-*Eucalyptus* terrestrial pollen grains were counted. Percentages were calculated using the sum of terrestrial pollen. Other aquatic pollen and spore percentages were calculated from a sum comprising of the terrestrial pollen and these palynomorphs. Stratigraphically constrained cluster analysis (CONISS) [96] was performed in Tilia 2.6.1 [97] and used to produce a dendrogram for the terrestrial pollen data. A broken stick model determined the number of significant pollen zones using the package rioja v. 0.9-15.1 [98] in R v.4.2.0 [93] (Figure 3). Detrended Correspondence Analysis (DCA) [99] was performed on percentage terrestrial pollen values using DECORANA in PC-ORD 6.08 [100] to examine compositional changes through time (Figure 4). The pollen data were square-root transformed prior to the DCA to downweigh over-represented taxa assumed in this Gaussian-based analysis.

### 2.3. Macroscopic Charcoal & Charanalysis

Macroscopic charcoal was sampled at 0.5 cm intervals for the entire core length according to standard protocols [101]. A 1.25 cm$^3$ sample was digested with 10% hydrogen peroxide ($H_2O_2$), sieved using 125 μm and 250 μm mesh diameter and counted under a microscope [101]. Microscopic charcoal was also counted during pollen identification. Charcoal particle size is a product of a range of factors including distance from fire, vegetation type, and fire intensity/severity [101–103]. Shifts in charcoal size fractions are often interpreted as changes in local versus distant source area (i.e., proximity) [101]. However, these shifts can also reflect changes in fuel biomass (i.e., vegetation) [104–106]. Fletcher et al. (2014, 2018) [104,105] argue that macroscopic charcoal, in their analysis of Tasmanian forest-fire dynamics, was biased toward burning of woody fuels. To understand broad changes in fire activity irrespective of charcoal particle size and/or morphology, time-series statistical analyses were performed using CharAnalysis 1.1 [107], which interpolates charcoal counts to the median sample resolution to produce equally spaced intervals and calculating charcoal accumulation rates (CHAR, particles cm$^{-2}$ yr$^{-1}$) using the results of the age-depth modelling. The process allows the identification of charcoal peaks, a statistically robust proxy for local fire episodes [108]. Charcoal peaks are identified as the positive residuals exceeding the 95th percentile threshold of a locally fitted Gaussian mixture CHAR

background model to the CHAR data (smoothed to 100 years). Accumulation rate data were calculated for microscopic charcoal in Tilia 2.0.37 [97] using an exotic *Lycopodium* spike added to samples during pollen processing.

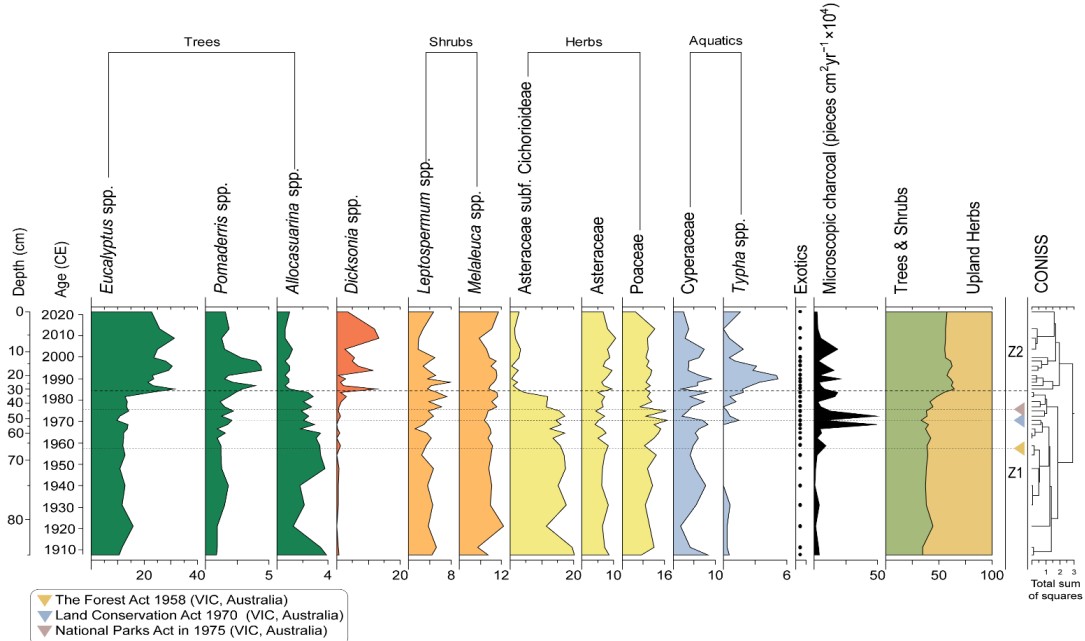

**Figure 3.** Terrestrial pollen stratigraphy by depth (cm) and age (CE) including microscopic charcoal accumulation calculated by known inputs of exotic *Lycopodium* spores. Two significant zones were determined (Z1 & Z2) using CONISS on the terrestrial pollen data. Here, dominant forest trees are represented in dark green, disturbance indicator, *Dicksonia* spp. in orange, dominant shrubs in orange, dominant herbs in yellow, abundant aquatic plants in blue and exotics as presence shown as black dots. Terrestrial pollen abundance was summed into two groups: trees & shrubs (light green) and upland herbs (gold). For interpretation of the references to colour in this figure legend, the reader is referred to the web version of this article.

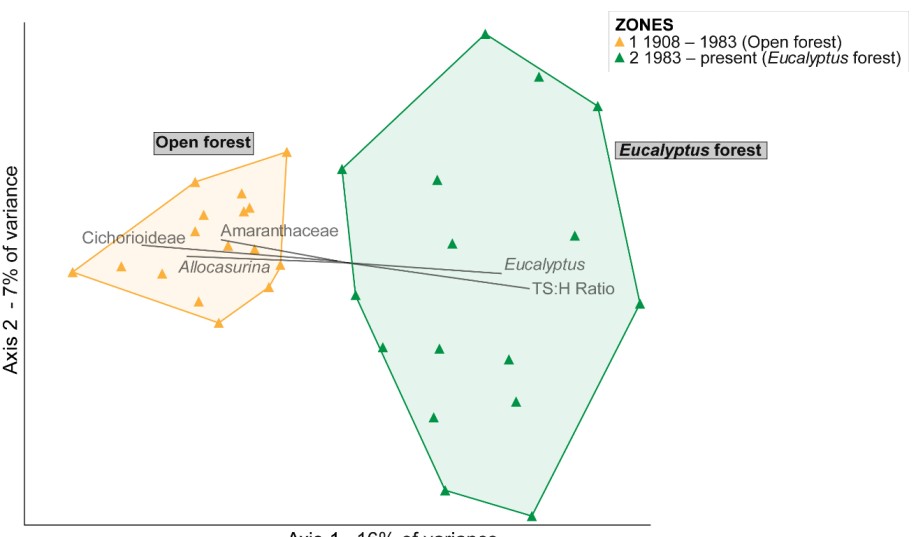

**Figure 4.** DCA ordination biplot showing the direction of correlation (cut-off $r^2$ value = 0.5) of pollen taxa with the ordination axes. Samples are grouped according to CONISS zones and naming follows the characteristic taxa that define each zone. For interpretation of the references to colour in this figure legend, the reader is referred to the web version of this article.

*2.4. Magnetic Susceptibility*

Geochemical analysis provides important information about post-fire erosion history and biochemical shifts [109,110]. The core was split with a Geotek core splitter [111] and a low field magnetic susceptibility profile was constructed using a Bartington MS2 m with MS2C sensor [112] at the Australian Nuclear Science and Technology Organization (ANSTO) at 0.5 cm intervals. This detects the abundance of and changes in ferrimagnetic minerals, which was used as an indication of changing erosional input to the lake [113–116].

*2.5. Numerical Data Analysis*

Generalised Additive Models (GAMs)

Generalised additive models (GAMs) were used to estimate trends in our unevenly spaced time-series data using smooth functions. GAMs use a sum of smooth functions to model non-linear trends, provide estimates of the magnitude of change and allow for the identification of periods of change [117–120]. GAMs were fitted to CHAR and magnetic susceptibility using mgcv v.1.8–25 [121] in R v.4.2.0 with the residual maximum likelihood (REML) method to penalise for overfitting trends [121,122]. A location-scale Gaussian GAM, which enables simultaneous estimation of both the mean and variance of a time-series, was used [117,123]. Base functions ($k$) were selected depending on the number of samples in the time-series being modelled to achieve the best model fit. Model prediction error was minimised by using Akaike's information criterion (AIC) [124]. Variance is frequently investigated in palaeoecological studies to identify critical transitions (when a threshold is crossed and the system shifts into a new stable state) of ecosystems [117,119,123,125–127]. Recent studies have highlighted that the variability of system behaviour may change in advance of a regime shift [117,123,128]. Therefore, increased variance is an important indicator of regime shifts. In this study, the implicit nonconstant variance (which arises due to each sample representing a different amount of time) was included as a covariate in the linear predictor for the variance part of the model [121,129].

## 3. Results

*3.1. Chronology*

A summary of the 210Pb results is presented in Table 1. The age model shows linear sedimentation (Figure 5a). Unsupported 210Pb activity reached background at 85 cm.

*3.2. Pollen*

We observe two significant CONISS zones from the percent terrestrial pollen taxa: Zone 1 (85–33 cm; ca. 1918–1983 CE) and Zone 2 (33–0 cm; ca. 1983 CE–present) (Figure 3). Zone 1 is dominated by herbs and grasses, predominately Asteraceae subf. Cichorioideae (20.3%) and Poaceae (17.2%). *Eucalyptus* spp. (15.9%) remain relatively low and overall, trees and shrubs (<50%) are of lower proportion throughout this zone. Zone 2 presents a doubling of *Eucalytpus* spp. (31.4%) and an increase in other tree and shrub taxa (64.4%), which remain dominant throughout this zone. *Dicksonia* spp. (13.4%) and *Typha* spp. (5.1%) become more prominent in this zone. Asteraceae subf. Cichorioideae (4.5%) and Allocasuarina (1.2%) sharply decrease and remain low (Figure 3). Exotics (*Pinus* spp., *Rumex* spp. and *Plantago* spp.) were present throughout the record. The terrestrial pollen DCA has two significant axes, with a 16% explained variance for axis 1 and 7% for axis 2 (Figure 4). DCA axis 1 has a strong negative correlation with Asteraceae subf. Cichorioideae (−0.981), *Allocasuarina* spp. (−0.843) and Amaranthaceae (−0.830) and a strong positive correlation with Trees & Shrubs:Herbs ratio (0.985) and *Eucalyptus* spp. (0.942) (Figure 4).

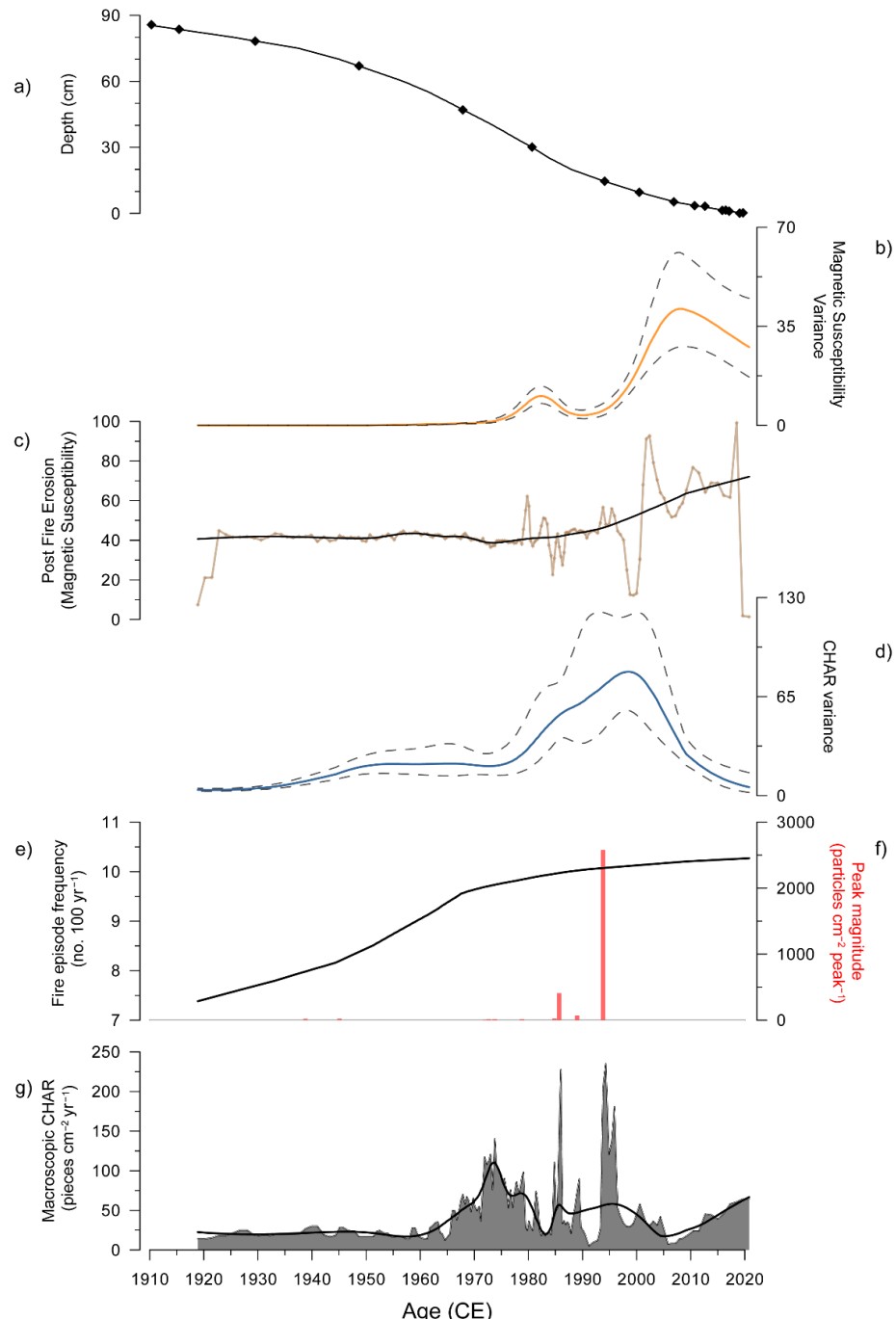

**Figure 5.** Results (**a**) Age-depth model (black line) with dating control points marked as black diamonds; (**b**) Magnetic susceptibility generalized additive model (GAM) fitted variance (orange line) and confidence intervals (grey dashed lines); (**c**) The magnetic susceptibility profile from core VIC2104A (brown line) with a GAM (black line); (**d**) CHAR GAM fitted variance (blue line) and confidence intervals (grey dashed lines); (**e**) Fire episode frequency (black line) (window-width = 100 years; note this is the minimum window width applicable) (CharAnalysis); (**f**) Peak magnitude (particles $cm^{-2}$ $peak^{-1}$), which measures fire size and intensity (red bars) (CharAnalysis); (**g**) Macroscopic charcoal accumulation rate (Macroscopic CHAR, pieces $cm^{-2}$ $yr^{-1}$) (grey) (window-width = 100 years) with a GAM (black line). For interpretation of the references to colour in this figure legend, the reader is referred to the web version of this article.

### 3.3. Macroscopic Charcoal & Charanalysis

CHAR is generally low from 86 to 65 cm (ca. 1918–1960 CE), followed by higher CHAR upwards from 65 cm (ca. 1960 CE to present) with maximum peaks occurring at 26 and 12 cm (ca. 1987 and 2000 CE) (Figure 5g). Fire episode frequency also begins to increase at 68 cm (ca. 1960 CE). The 2000 CE peak represents the largest magnitude fire episode reaching >2500 cm$^{-2}$ peak$^{-1}$ (Figure 5e,f).

### 3.4. Magnetic Susceptibility

Magnetic susceptibility is generally stable from ca. 1918 until ca. 1980, where it begins to increase and become variable (Figure 5c). Sharp increases and peaks occur in ca. 2000 and ca. 2019, with an overall higher trend within that period (Figure 5c).

### 3.5. Numerical Data Analysis

Generalised Additive Models

The CHAR GAM increases from ca. 1960 to a peak ca. 1974 then declines until an increase and overall stable values from ca. 1983 to present (Figure 5g). The variance of the CHAR residuals shows an increasing trend from ca. 1975 peaking at ca. 2000 and declining to present (Figure 5d). The magnetic susceptibility GAM begins an increasing trend from ca. 1980 to present, with the highest peak in the present (Figure 5c). The magnetic susceptibility variance starts to increase at a similar time ca. 1980 and sharply increases from ca. 1995 to a peak in ca. 2007 with a steady decrease to present (Figure 5b).

## 4. Discussion

### 4.1. Landscape Change between ca. 1900–2021

Our results highlight the effect that shifting cultural perceptions of the relationship between people and place had on the landscape around Buchan. Dense and fire-prone eucalypt-forest now dominates much of the landscape in the Buchan region. This dominance by dense flammable forest occurred in direct response to the banning of settler mimicry burning of this part of Gunaikurnai Country in the 1970s under legislation built on the central notion that landscapes needed protection from people. Prior to forest dominance, our empirical data suggest that the Buchan region was a herb and grass-rich open forest environment (Figures 3, 4 and 6a). Our data confirm the accuracy of the open forest scene depicted in the 1867 lithograph by Eugene von Guerard (Figure 1f). This open forest landscape consisted mostly of Asteraceae subf. Cichorioideae (Figures 3 and 4). While possibly representing species no longer local to the area, of the four local native Cichorioideae in the Buchan region [130], *Microseris lanceolata* (yam daisy), is among the most important economic plants for southeast Australian Aboriginal people [49,131–133]. It is possible that this site represents a cultivated yam field maintained by Gunaikurnai, although more analysis on the pollen taxonomy of this group is required to test this notion. Nevertheless, the grass and herb-dominant pollen spectra through this period reflect an open landscape between ca. 1900–1980 in which fire-sensitive trees (*Allocasuarina* spp.) were more common than today (Figure 3).

We identify a shift in fire activity in the late 1960s to early 1970s (age-depth model error range), from relatively low macroscopic charcoal input to both an increase in the amount of charcoal into the site and an increase in the variability of charcoal input. These changes reflect both an increase in the amount of burning of woody fuels (which produce larger charcoal fragments [101]) and a shift to a more variable fire regime (Figure 6c,f). This is followed by an increase in the proportion of vegetation comprised of woody fuels (trees and shrubs) in the late 1970s (Figure 6a) and a further increase in the variability of burning, with higher peaks and lower troughs in charcoal deposition (Figure 6e,f). Indeed, the most severe burning events recorded at the site occur following a doubling of tree and shrub pollen in the sequence, signalling significant increases in vegetation density (Figure 6). These events were severe enough to completely incinerate vegetation cover and expose soils to removal by erosion [106,116,134,135] (Figure 6a,b). In sum, the data

presented display a shift from a consistent high frequency-low intensity fire regime (i.e., cultural burning/settler mimicry) within an open grass and herb-rich forest, to a lower frequency-higher intensity fire regime (i.e., infrequent catastrophic fires) (Figure 6d) in response to the banning of settler mimicry burning under the Land Conservation Act of 1970—a wilderness-inspired act of legislation aimed at promoting and protecting people-free nature. The absence of a discrete charcoal peak in the upper most section of the sediment core reflecting the early 2020 "Black Summer" bushfires that burnt the region likely reflects the time-averaged nature of charcoal deposition into wetland sediments [101]. This results in a large part of the charcoal signal delivered into wetlands being washed in during the years subsequent to discrete fire events.

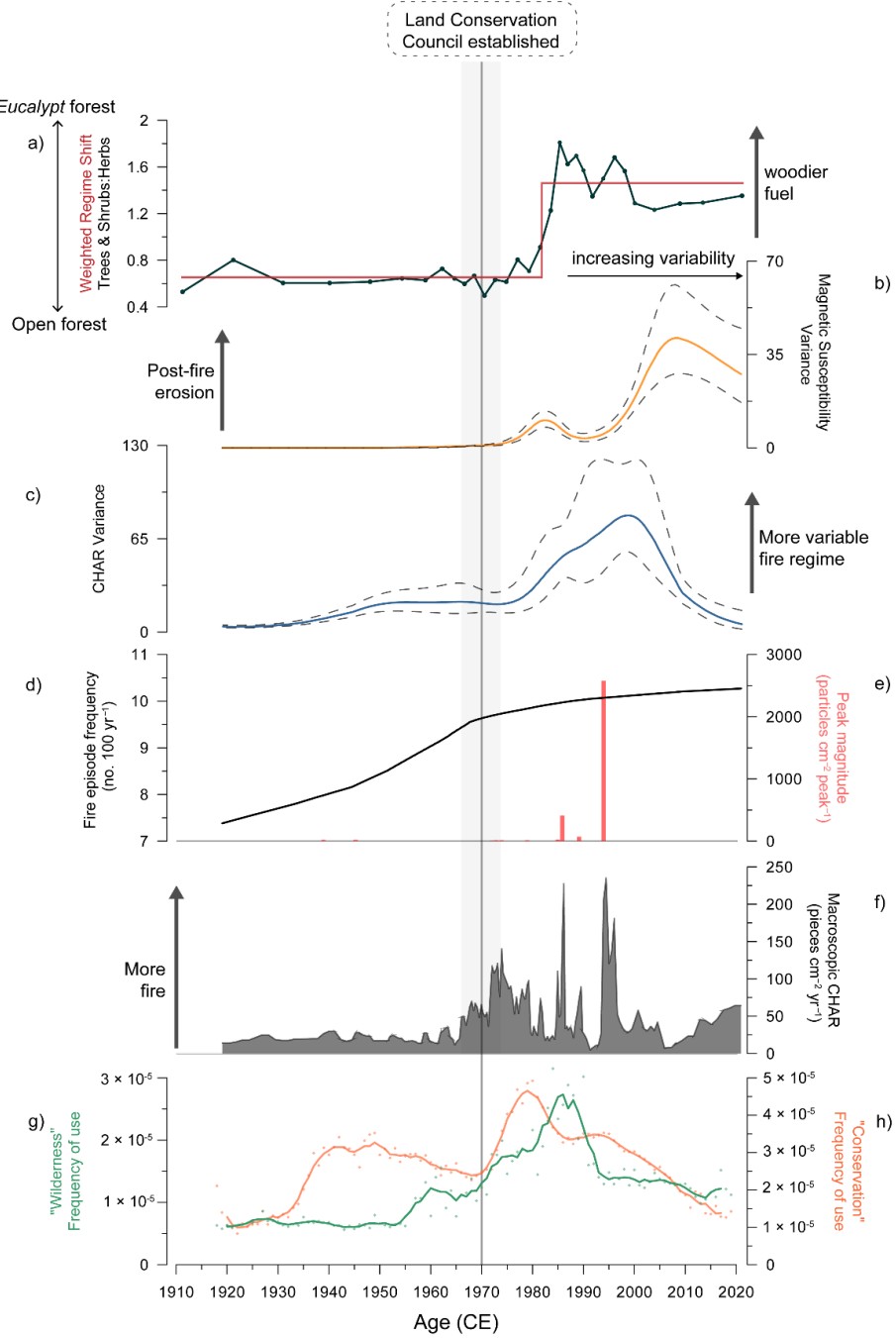

**Figure 6.** A summary plot of Tooculerdoyung Lagoon data (**a**) Percent ratio between Tree & Shrub taxa and Herb taxa (green line) with weighted regime shift (red line). The Rodionov sequential regime

shift detection method [136] represents successive applications of student's *t*-test to determine if the addition of a new observation to a set of *L* observations significantly ($p = 0.05$) changes the mean value of the time series, i.e., regime shifts; (**b**) Magnetic susceptibility generalized additive model (GAM) fitted variance (orange line) and confidence intervals (grey dashed lines); (**c**) CHAR GAM fitted variance (blue line) and confidence intervals (grey dashed lines); (**d**) Fire episode frequency (black line) (window-width = 100 years); (**e**) Peak magnitude (particles $cm^{-2}$ $peak^{-1}$) (red bars); (**f**) Macroscopic charcoal accumulation rate (Macroscopic CHAR, pieces $cm^{-2}$ $yr^{-1}$); (**g**) 'Wilderness' frequency of use Google ngram data (green) from 1918; (**h**) 'Conservation' frequency of use Google ngram data (orange) from 1918. The vertical grey line represents the establishment of the Victorian Land Conservation Council under the Land Conservation Act of 1970. The grey shading vertical shading around the 1970 line represents the age-depth model uncertainty for 1970. For interpretation of the references to colour in this figure legend, the reader is referred to the web version of this article.

### 4.2. The Environmental Impact of Legislation

The results of our study reflect the implications of changing land management practices that have occurred at Buchan, and at a global scale, as a result of conservation legislation that did not sufficiently consider the role of people as managers and carers of the environment/Country. Frequent, low-intensity burns by the Gunaikurnai custodians and subsequently by settlers mimicking Aboriginal practices discouraged the encroachment of shrub and tree-dense eucalypt-forests in favour of open grass and herb-rich woodlands surrounding our study site (Figure 6). Aboriginal people across what is now known as the Australian continent use burning as part of a suite of practices to care for Country [46,49,50,137]. The timing and frequency of fire is based on reciprocal and intimate relationships with Country and governed by strict cultural protocols and Law that are specific to place [39]. These intimate relationships with Country have developed over tens of thousands of years and through the course of enormous environmental shifts (such as glacial-interglacial cycles), shaping the biota and contributing to the configuration of Australian landscapes [41,44,104,116,138,139]. The failure to acknowledge or recognise the careful curation of Country by Aboriginal societies underpinned the concept of *terra nullius* that, under British law, justified the confiscation of Aboriginal lands by the British crown [14]. The surge in the use of the term "wilderness" in concert with the Invasion and subsequent colonisation of Australia, while possibly not causally related, is consistent with encountering new and foreign landscapes and mistaking the lack of familiar (to the invader) and easily identifiable indicators of agriculture (such as scythes, fences and ploughs) as an indication of a lack of human influence. The abundance of ethnohistoric data that describes a clear understanding by settlers of the power and importance of cultural burning by Aboriginal people suggests that this ignorance was willful rather than naive. Irrespective of intent, that narrative that Australia was and still is a vast wilderness has continued to the present day and has to a large extent shaped the regulatory framework around how this continent is managed [16].

The direct implication of wilderness-inspired conservation legislation on the local environment around Buchan on Gunaikurnai Country was profound. Prior to ca. 1970, biomass (i.e., fuel load) was predominantly herbs and grasses that fostered a broadly stable low-intensity fire regime (Figure 6a,d). Local Gippsland settlers state that their intent for frequently firing the land around Buchan was to mimic the management style they witnessed Aboriginal people practicing [47]. These settler mimicry fires were often different in frequency, intensity and seasons of burning, yet they were still able to maintain some semblance of pre-existing fire regimes (Figure 6). Buchan locals made note that the Gunaikurnai enacted frequent, cool, slow-burning fires that maintained open vegetation and low biomass [87]. In a submission to the Inquiry into the impact of public land management practices on bushfires in Victoria Hearing (2008), several stakeholders local to the Buchan region submitted the following statement as evidence for the role of regulation in driving an increase in catastrophic bushfires:

*"In earliest times the indigenous people used the firestick as a management tool—burning the dry grass, keeping the grassy areas fresh and green and ensuring a plentiful supply of wildlife. These fires were not dangerous—just slow burning and maintaining a balance in the bush. After the limitations on these people, the settlers followed their ways and the country retained its grassland quality. Forestry Officers took over this responsibility of maintaining a balanced public land service. These men had a good understanding of the bush and did a very good management job. Then the regulations began to be more and more restrictive. Public land management and the responsibility of the Minister, have been evaded, over a long time."* [87].

Despite understanding the clear benefits of maintaining landscape openness via fire, regulation increasingly restricted the mimicry of cultural burning as settlers began to fear fire around their assets [47]. It was at a similar time that the global conservation movement increasingly adopted the wilderness narrative to promote the exclusion of human activity from areas prized for their biodiversity and other natural assets [7,8,20,71]. In Victoria, from the late-1950s through to the mid-1970s the government set to create and enforce legislation as part of a conservation movement (Figure 2) that was not aimed at fuel reduction or fire prevention (largely ignoring fuel reduction), but rather fire suppression [140]. The direct impacts of these Acts can be seen in our data from Buchan, with: (1) a change in fire activity to more variable and severe fires; (2) a marked increase in woody and more flammable plants favoured by a lengthening of the fire-return interval and; (3) the onset of post-fire erosion events commencing after ca. 1970 and the establishment of the Land Conservation Council that implemented the prohibition of settler mimicry burning locally around Buchan and more broadly across Victoria (Figure 6) [76,89].

*4.3. The Curse of Conservation That Ignores People as Managers and History as a Prelude*

Climate change continues to dominate the narratives around the increase in the occurrence and intensity of catastrophic bushfires experienced in southeast Australia over the recent decades [34,52,141]. While it is beyond doubt that recent climate change produces conditions in which highly flammable forest fuels become easily combustible [142–145], the root cause of this trend is more complex than climate alone. An over-emphasis on climate runs the risk of developing mitigation strategies that fail to address the factors responsible for conditioning the landscape to burn under extreme fire weather [34]. Our data demonstrate that the change in approach to land management in the local area after the establishment of the Land Conservation Council in 1970 had a profound impact on the local region (Figure 6). The prohibition of burning by settler landowners imposed by the Land Conservation Council in the Buchan region [47,76,77] drove a shift from deliberate low-intensity burning of predominantly herbs and grasses that further promoted vegetation palatable for livestock to a climate-driven catastrophic fire regime dictated by the accumulation of forest fuel loads and their moisture content (Figure 6). The potential role that climate change has had on catastrophic bushfires in the study region cannot be discounted, with an increase in fire-promoting weather since 1950 in southeast Australia [146]. However, the timing and sequence of empirical changes we have observed (Figure 6), along with the local knowledge presented in bushfire enquiries across Gippsland, Victoria [47,140], implicate land-use legislation changes as the root cause for catastrophic bushfires in the Buchan region, and likely across southeast Australia. The accumulation of woody fuels around our study site, and, it is our contention, across the forested estate of much of southeast Australia today [41], follows the trajectory outlined in the schematic presented in Figure 7.

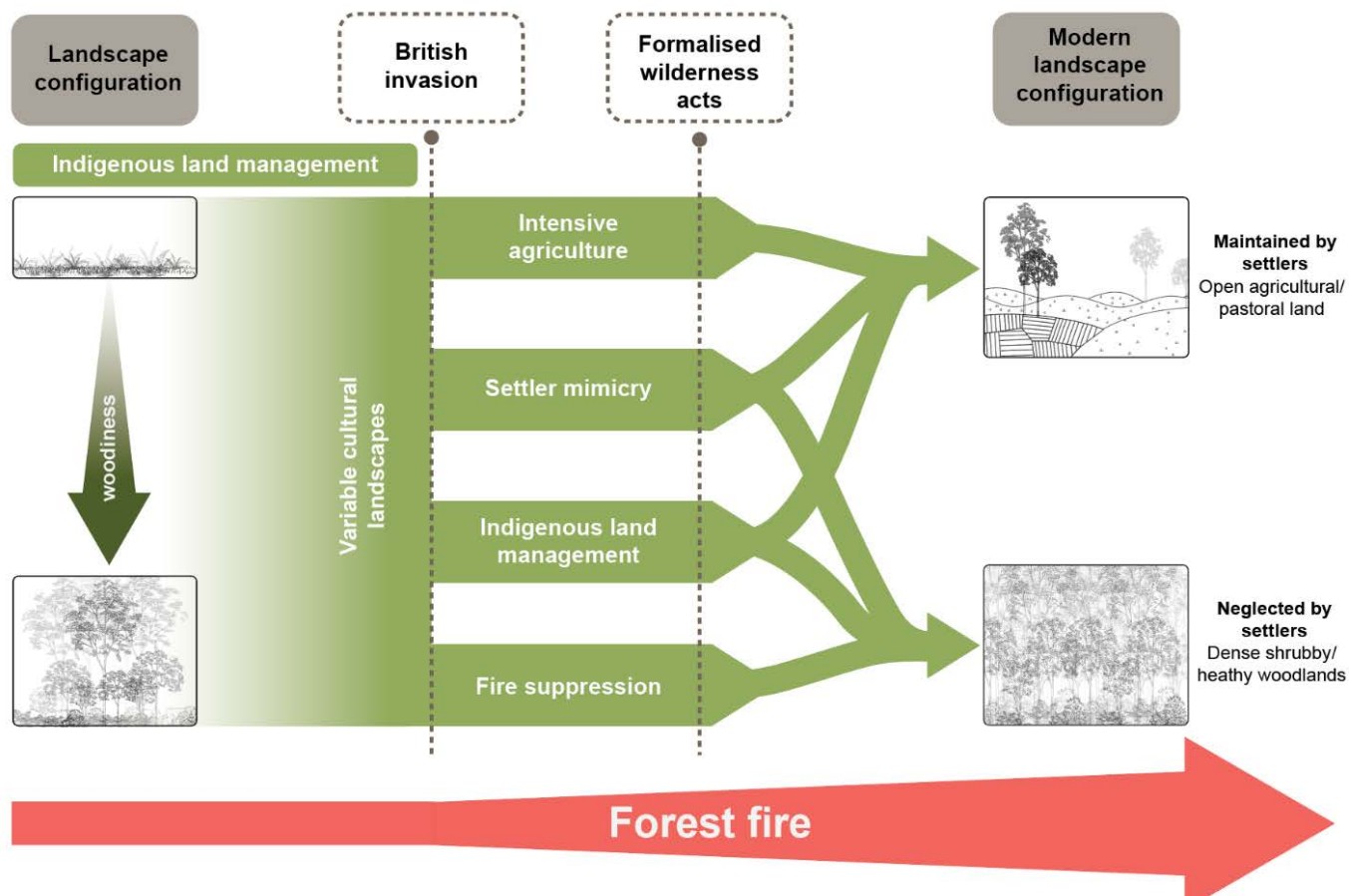

**Figure 7.** Conceptual diagram of how the landscape configuration and bushfire activity of southeast Australia has changed through time in response to the British Invasion and subsequent wilderness-inspired conservation legislation. Prior to British Invasion, Indigenous people managed variable cultural landscapes with fire across all parts of the region from open grasslands to closed forests in southeast Australia according to cultural protocols and economic objectives (left). After British Invasion management of landscape outside urban areas was fractured into four main approaches: (1) Intensive agriculture (taking advantage of open, grassy landscapes created and managed by Indigenous people, settlers concentrated agricultural activity and opened Country further); (2) Settler mimicry (in less open areas settlers mimicked cultural burning due to the recognition that Aboriginal managed landscapes were safer and more productive); (3) Indigenous land management (as removal and suppression of cultural burning was not uniform temporally and spatially, some parts of the landscape remained under Aboriginal management) and (4) Fire suppression (areas considered unsuitable for agriculture and/or settlement were neglected and fire suppression practices put in place) (center left) [34]. From the late 1950s legislation and conservation rooted in wilderness ideology further removed and suppressed active management of landscapes by Aboriginal and non-Aboriginal people (center right) facilitating a bifurcation of the landscape in to (1) open areas maintained by settlers as open pastoral or agricultural land and (2) neglected areas unmaintained by settlers as forest reserves, national parks and other areas where land was allowed to transition in to shrub-rich, dense woodlands in the absence of fire management (right) [41,75–77]. This confluence of factors has resulted in larger bushfires across southeast Australia since British Invasion (red arrow) [41]. For interpretation of the references to colour in this figure legend, the reader is referred to the web version of this article.

Figure 7 describes the pathway to the essentially bifurcated landscape (open agriculture land and closed forest) that characterises much of southeast Australia today. Open agricultural/pastoral land are maintained as open by mechanical intervention and/or

livestock grazing pressure—much of which was established on Country that was kept open by cultural burning prior to the British Invasion [48,49]. In places like our study area, this trajectory is marked by: (1) The removal of Aboriginal cultural burning following the British Invasion; (2) The replacement of cultural burning by management with fire aimed at mimicking Gunaikurnai management [47,87]; (3) A shift to an approach focused on the minimisation of human impact, in concert with the fire suppression ethos that governed forest management at the time (as it still largely does today) [75–77,87]; which resulted in (4) a landscape switch from low flammability open forest to high flammability closed forest (Figures 4 and 6a). This cultural shift in the way people perceived their role in the world around them from one of care and reciprocity to one of neglect and wilderness-inspired fortress conservation, preconditioned the landscape to be prone to climate-driven catastrophic bushfires.

The imposition of conservation policies that prevented controlled burning have resulted in the long-term neglect of Gunaikurnai Country. Gunaikurnai and other Aboriginal people cared for and curated healthy and biodiverse Country. This biodiversity, which is now being lost, was a direct product of the careful curation and care of Country by Aboriginal people. This loss is occurring because of the destructive impact of the predominantly extractive relationship that settlers have with this continent, compounded by decades of wilderness-inspired conservation and legislation designed to lock people out of Country. These extreme positions of how humans engage with the world around them is at the very heart of the environmental tragedies that grip the planet today. Denial of our place in Country fails to acknowledge our responsibility to care for Country. It paves the way for a destructive relationship with Country characterised by either abuse or neglect. The neglect of Country has created the conditions in which forests have become not only more flammable, but more susceptible to climate-driven bushfires [34,41]. Although there are moves to recognise the need for appropriate care for Country by Aboriginal people, such as co-management of several National Parks between Gunaikurnai and government agencies, neglect of Country pervades across much of the southeast Australian forest estate today. The resultant contiguity and high biomass load of the modern forest estate across southeast Australia represents a catastrophic risk to Country and reflects monumental land management failure on behalf of settler society in Australia. This is a failure borne from the willful ignorance of Aboriginal people and the imposition of dehumanising ideas of wilderness [16]; it is a push that continues to this very day, where Country created and maintained by Aboriginal people is still mapped and managed by settler societies and scientists as wilderness or euphemisms like "ecologically intact" [18,116,138,139].

The curse of this naive and inappropriate conservation approach is evident in the current bushfire and extinction crises currently impacting parts of the globe, with our study demonstrating that that such approaches can create conditions in which flammable landscapes prone to catastrophic bushfires become more flammable and suffer the impacts of recurrent catastrophic fire [34,41,147,148]. Perpetuation of the same myths and paradigms will only worsen these crises. It is time to listen to and learn from Indigenous and local people, and to empower these communities to drive research and management agendas [16,34].

**Author Contributions:** Conceptualisation, M.-S.F. and A.L.; methodology, A.L., A.R. and P.S.G.; formal analysis, A.L., A.R. and P.S.G.; writing—original draft preparation, A.L., M.-S.F. and A.R.; writing—review and editing, A.L., M.-S.F., A.R., R.M., S.C., M.M. and S.Y.M.; visualisation, A.R. and M.-S.F.; supervision, M.-S.F.; project administration, M.-S.F.; funding acquisition, M.-S.F., S.C., M.M. and S.Y.M. All authors have read and agreed to the published version of the manuscript.

**Funding:** This research was funded by the Australian Research Council, grant number IN210100055.

**Data Availability Statement:** Data for this project will be made available upon request.

**Acknowledgments:** We acknowledge Indigenous and local people whose Country we live and work on and whose knowledge and practice underpin what we have assembled here. We thank the Gunaikurnai and the Gunaikurnai Land and Waters Aboriginal Corporation for their support and

allowing us to work on their lands. Likewise, we thank Parks Victoria for providing permits to work in the Snowy River National Park. We would also like to thank Harriet Magee, Caitlyn O'Shea, Sarah Cooley and Callum Simpson for their assistance in the field. We would like to acknowledge Kristen Beck for her assistance with general additive models. We thank Bruno David and three anonymous reviewers for comments on an early draft of this manuscript.

**Conflicts of Interest:** The authors declare no conflict of interest.

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
