# Peer review of "The Curse of Conservation: Empirical Evidence Demonstrating That Changes in Land-Use Legislation Drove Catastrophic Bushfires in Southeast Australia"

_fire, doi:10.3390/fire5060175_

Round 1

Reviewer 1 Report

Review report

The paper, The curse of conservation: empirical evidence demonstrating that changes in land-use legislation drove catastrophic bush fires in southeast Australia, uses pollen and charcoal data to investigate the history of fire management on Gunaikurnai Country, in Australia. This an interesting and compelling paper that will stimulate discussion in much-needed areas of academia, politics and land management.

Specific comments:

Authorship: I commend co-authorship between academics and the Aboriginal organisation

1.    Introduction (overall): this is fascinating, leads well into article

Line 56-58: this sentence is difficult to understand (double negative), suggest that you use something closer to the title of ref #17

2.    Materials and methods

Line 250: what date was the core collected? Does the data include the impacts of the 2019/20 bushfires? 

Line 252: should ‘analysed’ be ‘analyse’ ?  should ‘elders’ be capitalised as ‘Elders’?

3.    Results

Fig 3. : can the y axis be extended to make the figure larger? It is a little difficult to read due to the small size

Fig 6 f.: this seems to suggest a period, more recently (2000 onwards), of less erratic bushfire behaviour. Are the 2019/20 bushfires included in this figure, if so, why is there no peak in charcoal during this period? 

4.    Discussion

Lines 635-639: this sentence is too long and convoluted and needs to be clearer and more precise/broken down into multiple sentences.

Line 648: is ‘neglect’ the correct term? It would seem that ‘exclusion’ of Indigenous communities is more relevant, rather than wilful neglect on behalf of the communities who are no longer able to access their Country.

Fig 6f (if I have interpreted correctly) appears to lead into a more recent period with less erratic fire behaviour. The increase in forest fire represented in Fig 7 does not necessarily align with the charcoal records in 6f. 

How have the fire management policies of government agencies influenced the recent history of bushfires? With an increase in hazard reduction burning across forested areas following previous bushfire events, how is this addressed in the paper? Does this paper consider hazard reduction burning by government agencies another form of settler mimicry? The large budgets of government agencies to manage bushfire in forested areas in recent times, contradict the notion that these areas have been neglected. It appears that relevant government policies from 1958 - 1975 are included, however more recent government policies that may have influenced the results presented here, have been overlooked.

Lines 660-662: this is an excellent concluding sentence.

Overall, this paper presents a clear and focused message on the importance of decolonising land management and empowering Indigenous communities to care for their Country. This is an important message that needs to be shared widely. However, there is a concern that by oversimplifying the conclusions and overlooking some of the complex factors in play, the paper may be vulnerable to criticism. Some of the other factors that could be acknowledged or addressed in the paper include the changing government policies and resources related to bushfire management over time, and other potential causes of woody thickening, such as climate change. The limitations of sampling only one core sample, in one locality, should also be explicitly addressed. The results for one local area cannot be extrapolated to an entire state area. 

Author contributions: the role of the Gunaikurnai Land and Waters Corporation is not addressed in this section.

Acknowledgements: line 670, possibly delete apostrophe (or change to comma) in peoples’

References: many of the references with corporate authors appear to need editing    

Author Response

File attached

Reviewer 2 Report

This is a very important paper, and I congratulate the research team on their excellent work. This article uses physical science research to argue that wildfire risk and ecosystem degradation has increased due to conservation practices. I only have a few minor comments.

1. The terms Indigenous, Aboriginal, and First Peoples are used interchangeably (especially in the abstract). I'd chose one, as it's quite confusing for myself as an international reader - I'm not sure if the authors' are referring to different groups? For example, in Canada, Indigenous is the overarching term for three different Indigenous groups (First Nation, Metis, and Inuit).

2. The terms forest fire and bushfire are also used throughout. Again, as an international reader, I'm not sure if these are referring to different types of fire in an Australian context?

3. Line 77 - after country, there is a parenthesis that should be deleted. 

4. Line 137-138 - currently forested should be deleted, as it's a bit confusing. So the sentence would then read "Prior to the British Invasion, landscapes in southeast Australia...".

5. Some of the figures and graphs are a bit hard to read when printed in black and white, especially Figure 2. A dashed line may help.

6. Figure 7 - should climate change be added to this figure?

7. I would make lines 634-662 the conclusion.

8. I think this article is missing a brief mention (one or two sentences) of limitations of physical science methods like pollen and charcoal sampling related to cultural burning. Many cultural burns are very small scale and may not be 'caught' in typical paleofire records. As a practitioner myself, I know that often my neighbour can't even tell if we are doing a small scale burn let alone producing enough charcoal to be seen in a charcoal record. I think the data produced here is important but potentially an undercount of all the cultural fire that could have been occurring. There was a good article that came out on this in 2019 by Roos and colleagues that could be cited: Roos, Christopher I., Grant J. Williamson, and David M.J.S. Bowman 2019    Is Anthropogenic Pyrodiversity Invisible in Paleofire Records? Fire 2:42.  

9. Some of the citations in the reference list show up strange (for example 21, 54, 80, 86 (also missing access date), 105). Seems like it could be a citation database error on how government reports are cited. Hopefully these will be corrected during the proofing stage.

Author Response

File attached

Reviewer 3 Report

This is an impressive paper. It is also such important scholarship.

The authors have brought together a suite of evidence and arguments to overturn many fictions that underscore non-Indigenous land management in Australia (and elsewhere), and have done so expertly. The clarity and strength of the voices brought together is to be commended. It fills an important gap in the literature, by providing a scientific counterpoint to the ecological science versus natural hazard science debates.

I have no feedback on the natural science content. With the qualitative work, I have very little to add. I make only two minor suggestions.

line 199-222, you could include a sentence about reduced Gurnai-Kurnai capacity to manage vis a vis the scale of the land and segregation policies. Note, I am not sure you say the area size in this introductory section. 

line 652 "wilful ignorance of Aboriginal people and their expertise"

Author Response

File attached
